# *Pseudomonas aeruginosa* Infections Are Associated with Infection Recurrence in Arteriovenous Grafts Treated with Revision

**DOI:** 10.3390/medicina59071294

**Published:** 2023-07-13

**Authors:** Yuan-Hsi Tseng, Chien-Chao Lin, Min Yi Wong, Chih-Chen Kao, Ming-Shian Lu, Chu-Hsueh Lu, Yao-Kuang Huang

**Affiliations:** 1Division of Thoracic and Cardiovascular Surgery, New Taipei Municipal TuCheng Hospital, College of Medicine, Chang Gung University, Taoyuan 33302, Taiwan; 8802003@cgmh.org.tw; 2Division of Thoracic and Cardiovascular Surgery, Chiayi Chang Gung Memorial Hospital, College of Medicine, Chang Gung University, Taoyuan 33302, Taiwangsl3639000@gmail.com (C.-H.L.); 3Microbiology Research and Treatment Center, Chiayi Chang Gung Memorial Hospital, Puzi City 613, Taiwan

**Keywords:** arteriovenous graft infection, *Pseudomonas aeruginosa*, graft infection, vascular access, dialysis access

## Abstract

*Background and Objectives*: This study was conducted to investigate whether Pseudomonas aeruginosa (PA) infections of arteriovenous grafts (AVGs) recur more frequently than other bacterial infections following treatment with revision. *Materials and Methods*: Operative procedures, including total excision, subtotal excision, and revision, were performed on 60 patients to treat 65 AVG infections. Final outcomes were classified as no infection recurrence, infection recurrence, and death without prior recurrence. In the competing risk setting, the cumulative incidence was estimated using the cumulative incidence function and Gray’s test, and the associations between outcomes and different variables were estimated using a subdistribution hazard (SDH) model. *Results*: Comparing AVG infections with and without recurrence, PA infection was not associated with a higher risk of infection recurrence (*p =* 0.13); however, the first operative procedure type was associated with infection recurrence (*p =* 0.04). AVGs with PA infection were associated with a higher total number of surgical interventions (*p <* 0.05) than AVGs without PA infection. Regarding the cumulative incidences of outcomes, for AVGs treated with subtotal excision or revision, the cumulative incidence of recurrent infection was 3.3-fold higher for those with PA infection than without one year after the first surgery. However, when AVGs were treated with revision alone, the cumulative incidence was 4.1-fold. After excluding AVGs treated with total excision, the SDH model was applied, obtaining a hazard ratio for infection recurrence of 16.05 (*p =* 0.02) for AVGs with PA infection compared with AVGs without PA infection. No other variables were significantly associated with infection recurrence. *Conclusions*: For subtotal resection and revision, AVGs infected with PA had a higher recurrence rate than those infected with other species. However, revision surgery may aggravate the recurrence rate.

## 1. Introduction

Arteriovenous grafts (AVGs) account for >20% of arteriovenous accesses [1,2]. Although AVGs are not the first choice for long-term dialysis, they remain preferable to arteriovenous fistulas (AVFs) for patients who have already started catheter-based dialysis or those with a shorter life expectancy because AVGs have a shorter maturation period and lower primary failure rate. Previous studies have also confirmed that AVGs have shorter durations of dialysis catheter use and comparable or higher survival rates for elderly patients undergoing dialysis than AVFs [3,4,5,6]. However, AVGs have a higher incidence of infections. Although antibiotics are the first-line treatment, most patients still require surgical intervention [7].

The optimal treatment course for infected AVGs is total excision, which presents with the lowest risk of recurrent infection, whereas revision is considered a less ideal option due to the high rate of recurrent infection [8,9]. However, total excision can be challenging due to the development of severe adhesions, and total excision can lead to bleeding, nerve injury, and arterial stenosis [10,11]. In addition, the creation of an entirely new AVG or arteriovenous fistula typically prolongs the use of a tunneled dialysis catheter, which is associated with multiple comorbidities [12,13]. Revision therefore retains its value in clinical application when the infection is focal or segmental. Although past studies have repeatedly emphasized that revision results in significantly higher rates of recurrent infections, other risk factors associated with infection recurrence have rarely been investigated.

In infected AVGs, Pseudomonas aeruginosa (PA) bacteria is detected at a lower proportion than Gram-positive cocci in many studies [8,9,11]. However, the treatment of PA infections remains challenging. PA bacteria generate a strong biofilm, making them resistant to many antibiotics, and produce elastase and alkaline protease enzymes that impair the host immune response [14,15]. In addition, Kim et al. reported that the environment of the fascia, where AVGs are located, can enhance PA virulence [16]. Therefore, AVG wounds infected with PA are likely to be very difficult to eliminate.

This study was conducted to investigate whether PA infection is a risk factor associated with infection recurrence in patients with AVGs who receive revision treatment.

## 2. Materials and Methods

This is a retrospective study. The inclusion criterion was that the AVG infection surgery was the first surgery, and the exclusion criterion was that the AVG infection surgery was not the first surgery. Indications for AVG placement are that the patient requires chronic dialysis. Among the 60 patients enrolled in this study, 65 AVGs underwent surgical intervention for AVG infection. In this study, infection recurrence is defined as a recurrent infection that occurs in the same AVG location as the first infection and is also caused by the same bacterial strain. Data were collected from September 2014 to November 2021, including age, sex, comorbidities, preoperative blood examination results, perioperative bacterial culture outcomes (including wound, pus, or blood cultures), the first operative procedure type (including total excision, subtotal excision, and revision), the number of days from suspicion of infection to the first surgery, the total number of surgical interventions received, recurrence-free follow-up time, and final clinical status. The follow-up period was until 31 November 2021 or the patient’s death.

Final outcomes were classified as no infection recurrence, infection recurrence, and death without prior recurrence. No infection recurrence was defined as patients who remained alive with no recurrence in AVG infection during the study period. Infection recurrence during the study period was the event of interest for this study. Because death without prior recurrence precludes the occurrence of infection recurrence, this outcome was defined as a competing event in this study.

The number of surgical interventions received for AVG infection was evaluated, excluding the following interventions: (1) incision and drainage alone, without graft excision, and (2) the changing of wound sealing dressing kits only. Total excision, subtotal excision, and revision were included. Total excision was defined as the removal of the entire graft with no remnant, in which the artery and vein are repaired directly or with a vein patch. Subtotal excision was defined as the removal of the infected graft segment, in which graft stumps are retained on the artery, the vein, or both. Revision, or the segmental bypass technique, was defined as the removal of the infected graft segment and the creation of a new subcutaneous tunnel using a new graft segment to connect the bilateral stumps for the restoration of AVG blood flow.

Total lymphocyte count (TLC) was calculated to evaluate the nutritional status of patients. In previous studies, a TLC of less than 900/µL was considered indicative of severe malnutrition, and a decrease in TLC has been associated with an increased mortality rate [17,18]. A TLC was necessary, as only a very small number of patients routinely receive preoperative nutritional assessments, such as assessing blood albumin levels.

At our hospital, we use the GORE INTERING Vascular Graft for both new AVG placements and revisions.

### Statistical Analyses

Categorical variables are summarized as the number and percentage, and continuous or ordinary variables are presented as the median and interquartile range. In the univariate analysis, a Chi-square test or Fisher’s exact test was used to analyze categorical variables, and continuous variables were compared using a Mann–Whitney U test. Due to the small sample size enrolled in this study, all variables with *p*-values less than 0.2 in the univariate analysis were included in the adjusted hazard regression model to estimate the effects of covariates.

In the case of the existence of a competing event (death without prior recurrence), the crude cumulative incidences for infection recurrence and death without prior recurrence were estimated with cumulative incidence functions (CIFs), and Gray’s test was used to compare the CIFs. In addition, because the majority of infection recurrence events occur within 1 year of the original infection, the prediction of cumulative incidences and the presentation of cumulative incidence curves were truncated at 1 year to improve readability.

Associations between variables and infection recurrence or death without prior recurrence were assessed by calculating the hazard ratios (HR) and 95% confidence interval using the subdistribution hazard (SDH) model. In addition, although the effect size of variables was large, achieving a *p*-value of < 0.05 in the univariate analysis was difficult because our sample size was very small. Therefore, we adjusted the threshold of the variables included in the SDH model to *p <* 0.2 to avoid missing potential confounding.

Statistical analyses were conducted using EZR (Version 1.55; Saitama Medical Center, Jichi Medical University).

## 3. Results

Of the infected AVGs that required surgical treatment, 15 had recurrent infections. The timing of the surgical intervention was when clinical symptoms, such as erythema, fluctuant or tender mass, purulence from a wound, graft material exposure, or skin necrosis, were identified when an AVG infection was present. Two patients experienced re-infection with different bacteria, one 5.2 years after revision and the other 2 years after revision. Both instances of infection were treated as distinct AVG infections, as infection with different bacteria is likely due to a new infection source rather than a recurrence of the original infection. The proportion of PA infections was higher in the recurrence group than in the non-recurrence group, but this difference was not significant (20% vs. 6%, *p =* 0.13). In addition, slightly more men were in the recurrence group than in the non-recurrence group but did not achieve significance (60% versus 30%, *p =* 0.07). No significant differences between the recurrence and non-recurrence groups were observed for patient comorbidities, blood exam results, or C-reactive protein levels. The first operative procedure type differed significantly between the recurrence and non-recurrence groups (*p =* 0.04), and total excision was associated with the lowest rate of recurrent AVG infections. The baseline demographics and clinical characteristics of the patients included in this study are presented in Table 1.

Of the six AVGs infected with PA, one patient suffered an infection at the access site and received a subtotal excision but died of sudden lower gastrointestinal bleeding on day 23 after the revision surgery. Two other patients who experienced bacteremia underwent total excision during the first surgery, and neither of these patients experienced a recurrent infection; however, one patient died on day 467 after the surgery due to refusal to undergo hemodialysis. The remaining three patients received revision surgeries, and all suffered recurrence of infection. Comparing AVGs with and without PA infection (Table 2), AVGs with PA infection had higher proportions of white blood cells (WBC) > 11000/µL (83.3% vs. 35.6%, *p =* 0.03) and greater total numbers of operations for both the AVG infection and recurrence (*p <* 0.05). In addition, the group with AVGs with PA infections had a slightly higher proportion of infection recurrence than those with AVGs infected by other bacterial species but did not reach significance (50% vs. 20.3%, *p =* 0.08).

Figure 1 shows the CIF estimation for infection recurrence and death without prior recurrence among AVGs with and without PA infection. At 1 year after the first surgery, the estimated cumulative incidence for infection recurrence was 58.3% for AVGs with PA infection and 17.9% for AVGs without PA infection. However, the CIFs for infection recurrence did not differ significantly between AVGs with and without PA infection (*p =* 0.07). In addition, the estimated cumulative incidence for death without prior recurrence was 16.7% for AVGs with PA infection and 17.5% for AVGs without PA infection, which was not significantly different (*p =* 0.88).

However, the group of patients who received total excision differed significantly from those who received subtotal excision and revision. Figure 2 shows that patients in the group treated with total excision did not experience infection recurrence, but the rate of death without prior recurrence was higher among these patients than among patients who received revisions (35.7% vs. 20.0%) or subtotal excisions (35.7% vs. 16.1%), although these differences were not significant (*p =* 0.44 and 0.24). In addition, the patients who underwent total excision had higher WBC counts (*p =* 0.04), band cell percentages (*p =* 0.01), and C-reactive protein levels (*p =* 0.03), and showed trends toward increased segmented neutrophil percentages (*p =* 0.06) and a higher proportion of extensive abscesses (*p =* 0.06) (Table 3). Patients who underwent total excision also had a lower proportion of lymphocytes than the non-total excision group (*p =* 0.04). We therefore excluded AVGs treated with total excision from further competing risk analyses.

In infected AVGs treated with subtotal excision or revision surgery (Figure 3), the estimated cumulative incidence for infection recurrence was 75% for AVGs with PA infection and 22.4% for AVGs without PA infection at 1 year post first surgery, which represents a significant difference (*p =* 0.03). However, the estimated cumulative incidence for death without prior recurrence was 25% for AVGs with PA infection and 12.6% for AVGs without PA infection, which was not significantly different (*p =* 0.19)

The individual analysis investigated AVGs treated with subtotal excision or revision alone. However, the only patient with an AVG PA infection treated with subtotal excision died within 23 days; therefore, we did not think that the CIF estimation would be meaningful. However, the CIF estimation for infected AVGs treated with revision (Figure 4) resulted in a cumulative incidence for infection recurrence of 100% for AVGs with PA infection and 24.6% for AVGs without PA infection at 1 year post first surgery, which was significantly different (*p =* 0.015). The estimated cumulative incidence for death without prior recurrence did not differ significantly between AVGs with and without PA infection (*p =* 0.67).

Because the risk of recurrent infection was almost zero with the total excision surgical approach in both previous studies and our results, we considered it meaningless to include total excision in the SDH model. Therefore, we excluded total excision, comparing only subtotal excision and revision. After excluding AVGs treated with total excision, the associations between outcomes (infection recurrence and death without prior recurrence) and variables identified as potentially impactful based on p-values less than 0.2 in the univariate analysis (Table 1) were investigated with the SDH model. The subdistribution HR for infection recurrence was 16.05 for AVGs with PA infection relative to AVGs without PA infection (*p <* 0.01; Table 4). In addition, band cell percentage was associated with the risk for death without prior recurrence (subdistribution HR, 2.07, *p <* 0.01). No other variables had significant effects on the cumulative incidence for infection recurrence or death without prior recurrence.

## 4. Discussion

AVGs are known to have higher rates of infection than autogenous arteriovenous fistulae. The majority of pathogenic bacteria identified in AVG infections are Gram-positive cocci, and the proportion of PA infections is reported to be < 15% in the majority of studies [8,9,19,20,21]. In some studies of AVG infections, PA infections are not described [22,23], likely because the primary cause of AVG infections is cannulation for dialysis [19,24], although some are also caused by intraoperative contamination. However, PA-associated AVG infections remain clinically important, as our study identified three patients who underwent AVG revisions and suffered from PA infection recurrence. Compared with Staphylococcus species, PA generates stronger biofilms that resist antibiotics and disturb cytokine function through the release of elastase and protease enzymes. PA may also display increased virulence when located in the fascia [14,15,16]. In addition, we believe that the low proportion of AVGs infected with PA results in delayed treatment because initial empirical antibiotic therapy is typically aimed at treating Gram-positive bacteria (using antibiotics such as oxacillin or vancomycin).

Recent studies have indicated that total excision is the ideal treatment option, and the early reconstruction of new hemodialysis access is also recommended [8,9,11]. However, in cases of localized infections without signs of systemic involvement, attempting to salvage arteriovenous access through the creation of a new subcutaneous tunnel and restore blood flow using a new graft segment remains a common surgical practice [25]. Although total excision is recommended for extensive infections, the treatment of serious adhesions requires a high level of technique and skill, including the ability to perform vein patches, vascular bypasses, or vessel ligations when necessary [26]. The creation of a new AVG is associated with infection risk but may prolong the ability to use a tunnel dialysis catheter. In our opinion, the identification of risk factors that may result in revision failure, including the effects of PA infection, is important for decision-making in cases of AVG infection.

Our study also revealed that despite a lack of infection recurrence in the total excision group, the mortality rate was higher than for the subtotal excision or revision groups. The high mortality rate may be associated with the occurrence of more serious infections, which tend to result in the decision to perform total excision. Compared with the group that received non-total excision, the group that received total excision had higher values for WBC count, band cell proportion, and C-reactive protein level, with a greater percentage of segmented neutrophils and an increased proportion of extensive abscesses (both *p =* 0.06). Clinically, these outcomes are reasonable, and previous studies recommend that vascular surgeons perform total excision in cases with extensive abscesses, particularly with the involvement of graft anastomosis or sepsis [10,21,27]. In our hospital, we observed that the extent of the abscess impacted the surgeon’s choice of intervention. Previous studies have also demonstrated that infection is a lethal event for dialysis patients [28,29]. Therefore, the early detection of AVG infection and adequate treatment is likely to reduce patient mortality and increase the revision success rate.

In designing our study, we chose not to consider incision and drainage alone (without partial or total graft excision) as a formal surgical intervention. In our experience, infection is nearly impossible to eradicate using this technique. Although previous studies have demonstrated that incision and drainage alone can be effective in certain patients, the technique is more commonly performed together with partial excision, with or without a new graft interposition, or with total excision [20,25,30,31]. In addition, a tissue flap can be used to cover the exposed AVG after the removal of damaged skin [32,33]; however, this approach is rarely used, and insufficient evidence exists to support a good prognosis using this technique. Despite this, we suggest that incision and drainage remains a useful technique when responding to the incidental discovery of AVG infection because it can be performed under local anesthesia and is well tolerated by most patients.

Previous studies and clinical experience have identified multiple strategies for reducing the rate of infection. The use of strict sterile technique in the surgical environment during arteriovenous access surgery and the administration of prophylactic antibiotics can reduce surgical site infection rates, although prophylactic antibiotic use remains controversial [34,35]. In addition to the aseptic technique, the buttonhole technique should not be used for arteriovenous cannulation due to the high rate of infection [36]. Despite the prevalence of these techniques, the infection rate for AVGs remains as high as 35% [26]. Regular AVG monitoring and surveillance are recommended for the early detection of significant stenosis and the identification of early graft infection, particularly because AVG infection is often complicated by bacteremia or remote infections that result in high mortality rates among dialysis patients [28,29,37].

This study had several limitations. First, the small sample size may have resulted in a degree of bias, and the differences between the three operative procedure groups were challenging to analyze in detail. Second, this study was performed at a single institution, which reduces the generalizability of our results. Third, the first operative procedure was influenced by the surgeon’s preference, and in some cases, total excision was performed even for localized infections, which may also have produced bias in our results. Despite these limitations, this study still attempted to investigate the association between AVG infection recurrence and the specific strain of bacteria involved, with a focus on PA infection.

## 5. Conclusions

Although limited data are available, our results suggest that, for subtotal resection and revision surgery, AVGs infected with P. aeruginosa had a higher recurrence rate than those infected with other species. However, revision surgery may aggravate the recurrence rate. In addition, additional data are required to determine whether there is a real difference in recurrence rates between subtotal resection and revision.

In addition, early intervention prior to clinical deterioration may reduce the necessity for total excision. Early intervention in AVG infections may, therefore, increase the revision success rate.

## Figures and Tables

**Figure 1 medicina-59-01294-f001:**
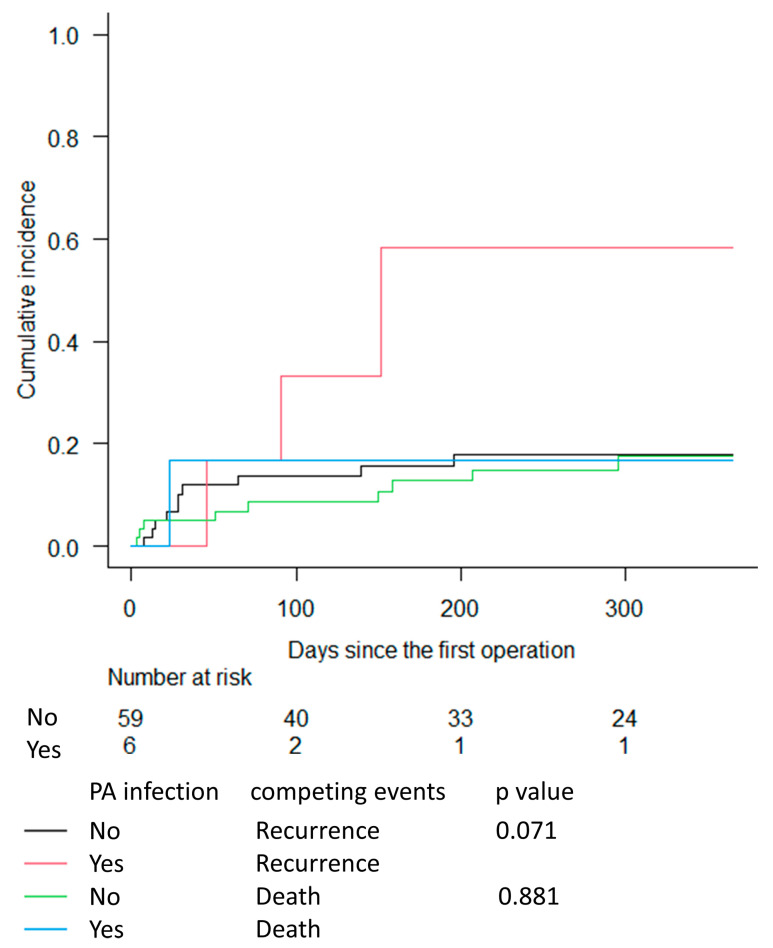
Cumulative incidence functions for infection recurrence and death without prior recurrence, grouped according to the presence or absence of PA infection, with follow-up time truncated at 1 year. Abbreviations: PA: Pseudomonas aeruginosa.

**Figure 2 medicina-59-01294-f002:**
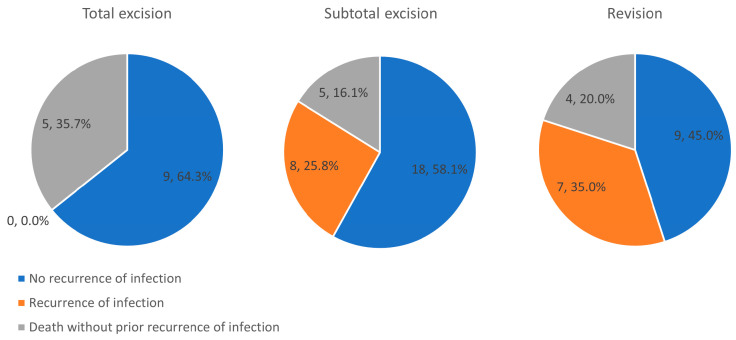
Pie chart showing the proportions of outcomes for the three operative procedures.

**Figure 3 medicina-59-01294-f003:**
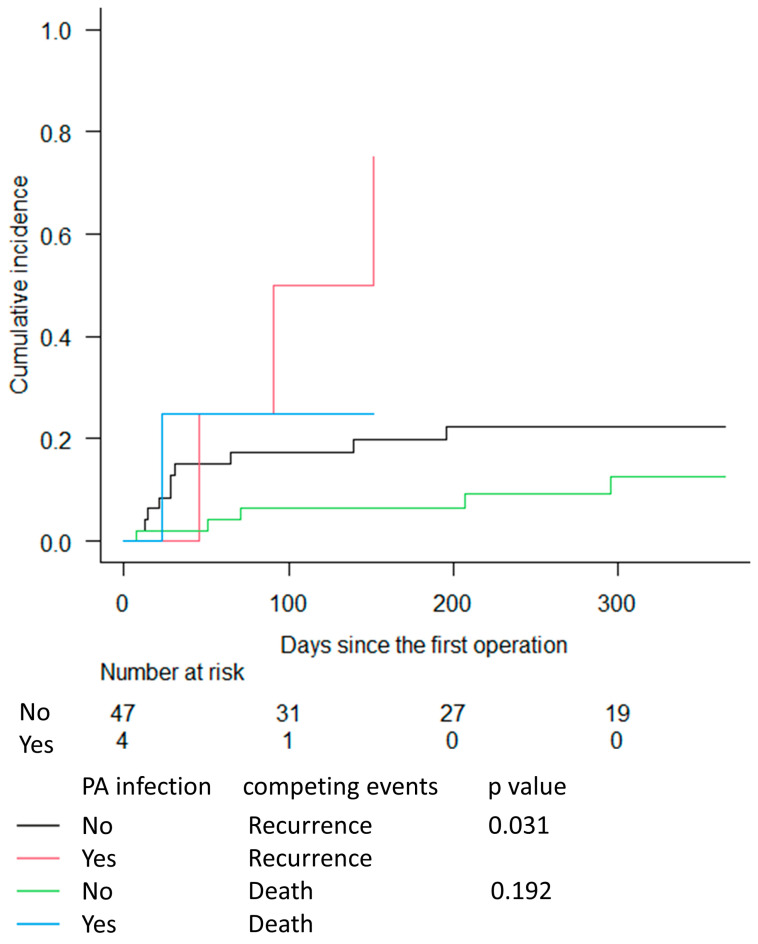
Cumulative incidence functions for infection recurrence and death without prior recurrence for AVGs treated with subtotal excision or revision, grouped according to the presence or absence of PA infection, with follow-up time truncated at 1 year. Abbreviations: AVG: arteriovenous graft; PA: Pseudomonas aeruginosa.

**Figure 4 medicina-59-01294-f004:**
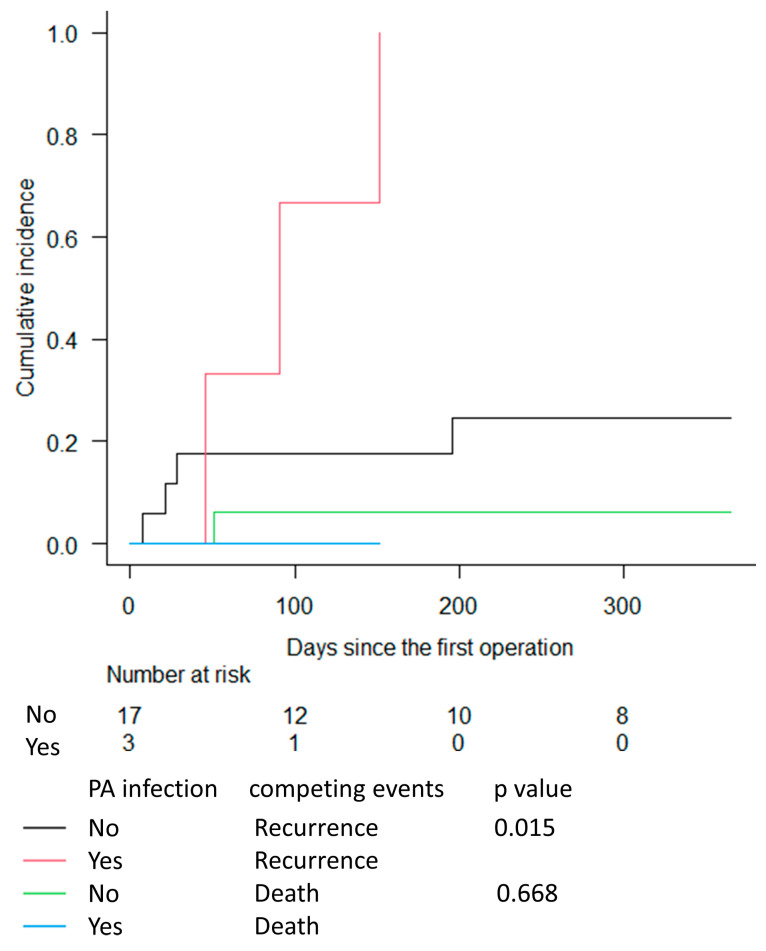
Cumulative incidence functions for infection recurrence and death without prior recurrence for AVGs treated with revision, grouped according to the presence or absence of PA infection, with follow-up time truncated at 1 year. Abbreviations: AVG: arteriovenous graft; PA: Pseudomonas aeruginosa.

**Table 1 medicina-59-01294-t001:** Baseline demographics and clinical characteristics of AVG infections in patients with and without infection recurrence following surgical intervention.

	No Infection Recurrence N = 50	Infection Recurrence N = 15	*p*-Value
Age (years)	68.50 [59.00, 78.00]	64.00 [59.50, 70.50]	0.30
Sex			0.07
Female, n (%)	35 (70.0)	6 (40.0)	
Male, n (%)	15 (30.0)	9 (60.0)	
Comorbidity			
DM, n (%)	36 (72.0)	11 (73.3)	1.00
HTN, n (%)	44 (88.0)	13 (86.7)	1.00
CAD, n (%)	12 (24.0)	5 (33.3)	0.51
EF < 40%, n (%)	3 (6.2)	1 (7.7)	1.00
History of CVA, n (%)	9 (18.0)	3 (20.0)	1.00
PAOD, n (%)	9 (18.0)	2 (13.3)	1.00
Smoking			0.29
Recent, n (%)	1 (2.0)	1 (6.7)	
Quit, n (%)	8 (16.0)	4 (26.7)	
Hyperlipidemia, n (%)	11 (22.0)	6 (40.0)	0.19
CLD, n (%)	19 (38.0)	5 (33.3)	1.00
Blood exam before 1st operation			
WBC (1000/µL)	9.90 [7.53, 12.70]	9.60 [6.60, 12.05]	0.48
WBC > 11,000/µL	21 (42.0)	5 (33.3)	0.77
Segmented neutrophil (%)	83.20 [74.20, 86.20]	77.10 [72.70, 83.00]	0.16
Band cell (%)	0.00 [0.00, 0.00]	0.00 [0.00, 0.00]	0.18
Lymphocyte (%)	9.10 [5.00, 14.25]	10.20 [6.90, 17.70]	0.57
TLC < 900/µL	21 (42.9)	6 (42.9)	1.00
CRP (mg/L)	42.00 [19.20, 115.98]	40.56 [12.52, 54.48]	0.37
Bacterial culture			
No growth, n (%)	2 (4.0)	1 (6.7)	0.55
PA, n (%)	3 (6.0)	3 (20.0)	0.13
OSSA, n (%)	24 (48.0)	5 (33.3)	0.38
MRSA, n (%)	17 (34.0)	4 (26.7)	0.76
Others, n (%)	12 (24.0)	6 (40.0)	0.32
Bacteremia, n (%)	36 (80.0)	10 (71.4)	0.49
First operative procedure			0.04
Total excision, n (%)	14 (28.0)	0 (0.0)	
Subtotal excision, n (%)	23 (46.0)	8 (53.3)	
Revision, n (%)	13 (26.0)	7 (46.7)	
Pus discharge, n (%)	29 (58.0)	8 (53.3)	0.77
Days from impression to 1st surgery, n (%)	5.00 [2.00, 9.75]	7.00 [4.00, 8.00]	0.37
Total number of operations, n (%)	-	2.00 [2.00, 2.50]	-

AVG: arteriovenous graft; CAD: coronary artery disease; CLD: chronic liver disease; CRP: C-reactive protein; CVA: cerebrovascular accident; DM: diabetes mellitus; HTN: hypertension; EF: ejection fraction; MRSA: methicillin-resistant Staphylococcus aureus; OSSA: oxacillin-susceptible Staphylococcus aureus; PA: Pseudomonas aeruginosa; PAOD: peripheral arterial occlusive disease; TLC: total lymphocyte cells; WBC: white blood cells.

**Table 2 medicina-59-01294-t002:** Comparison of characteristics between AVGs with and without *Pseudomonas aeruginosa* infections.

	No PA Infection N = 59	PA Infection N = 6	*p*-Value
Age (years)	68.00 [59.00, 77.00]	61.00 [57.00, 68.00]	0.32
Sex			0.40
Female, n (%)	36 (61.0)	5 (83.3)	
Male, n (%)	23 (39.0)	1 (16.7)	
Comorbidity			
DM, n (%)	43 (72.9)	4 (66.7)	0.67
HTN, n (%)	52 (88.1)	5 (83.3)	0.56
CAD, n (%)	17 (28.8)	0 (0.0)	0.33
EF < 40%, n (%)	4 (7.1)	0 (0.0)	1.00
History of CVA, n (%)	11 (18.6)	1 (16.7)	1.00
PAOD, n (%)	10 (16.9)	1 (16.7)	1.00
Smoking			1.00
Recent, n (%)	2 (3.4)	0 (0.0)	
quit, n (%)	11 (18.6)	1 (16.7)	
Hyperlipidemia, n (%)	14 (23.7)	3 (50.0)	0.18
CLD, n (%)	22 (37.3)	2 (33.3)	1.00
Blood exam before 1st operation			
WBC (1000/µL)	9.60 [7.25, 11.85]	13.35 [12.85, 13.85]	0.08
WBC > 11,000/µL	21 (35.6)	5 (83.3)	0.03
Segmented neutrophil (%)	79.60 [73.17, 86.00]	84.15 [83.23, 85.00]	0.61
Band cell (%)	0.00 [0.00, 0.00]	0.00 [0.00, 0.75]	0.40
Lymphocyte (%)	9.15 [6.08, 16.35]	9.10 [6.00, 9.95]	0.59
TLC < 900/µL	24 (42.1)	3 (50.0)	1.00
CRP (mg/L)	44.40 [19.36, 98.92]	10.27 [9.20, 119.70]	0.55
Bacteremia, n (%)	40 (75.5)	6 (100.0)	0.32
First operative procedures			0.25
Total excision, n (%)	12 (20.3)	2 (33.3)	
Subtotal excision, n (%)	30 (50.8)	1 (16.7)	
Revision, n (%)	17 (28.8)	3 (50.0)	
Total number of operations for recurrence of infection, n (%)	1.00 [1.00, 1.00]	1.50 [1.00, 2.75]	<0.05
Recurrence-free follow-up time (days)	266.00 [68.00, 609.50]	93.00 [57.25, 137.00]	0.18
Outcomes			0.08
No recurrence of infection, n (%)	35 (59.3)	1 (16.7)	
Recurrence of infection, n (%)	12 (20.3)	3 (50.0)	
Death without prior recurrence, n (%)	12 (20.3)	2 (33.3)	

AVG: arteriovenous graft; CAD: coronary artery disease; CLD: chronic liver disease; CRP: C-reactive protein; CVA: cerebrovascular accident; DM: diabetes mellitus; EF: ejection fraction; HTN: hypertension; PA: *Pseudomonas aeruginosa*; PAOD: peripheral arterial occlusive disease; TLC: total lymphocyte cells; WBC: white blood cells.

**Table 3 medicina-59-01294-t003:** Comparison of preoperative blood sample analyses and the extent of abscess between AVG infections treated with total excision and those treated with non-total excision.

	Non-Total Excision * N = 51	Total Excision N = 14	*p*-Value
Blood exam			
WBC (1000/µL)	9.60 [6.90, 11.75]	12.15 [8.85, 14.98]	0.04
Segmented neutrophil (%)	79.00 [72.55, 85.50]	84.00 [80.00, 91.00]	0.06
Band cell (%)	0.00 [0.00, 0.00]	0.00 [0.00, 2.30]	0.01
Lymphocyte	10.20 [7.10, 17.15]	7.00 [4.00, 9.20]	0.04
CRP (mg/L)	37.55 [14.66, 61.06]	132.33 [35.05, 199.81]	0.03
Bacteremia, n (%)	35 (76.1)	11 (84.6)	0.71
Abscess ^#^	N = 26	N = 11	0.06
Focal	12 (46.2)	1 (9.1)	
Extensive	14 (53.8)	10 (90.9)	

* Including subtotal excision and revision. ^#^ Based on surgical records. AVG: arteriovenous graft; CRP: C-reactive protein; WBC: white blood cells.

**Table 4 medicina-59-01294-t004:** Hazard ratio (and 95% confidence interval) obtained using the subdistribution hazard model for the recurrence of infection and death without prior recurrence for arteriovenous grafts treated with subtotal excision or revision.

	Infection Recurrence			Death without Prior Recurrence		
Variable *	HR ^#^	95% CI	*p*-Value	HR ^#^	95% CI	*p*-Value
PA infection	16.05	2.69–95.67	<0.01	0.64	0.18–2.26	0.49
First operative procedure	0.62	0.15–2.58	0.52	2.62	0.44–15.50	0.29
Sex	4.09	0.94–17.74	0.06	0.84	0.14–5.15	0.85
Hyperlipidemia	2.55	0.81–8.07	0.11	0.22	0.02–3.22	0.27
Band cell	0.64	0.41–1.01	0.05 ^†^	2.07	1.37–3.15	<0.01
Segmented neutrophil	1.00	0.97–1.03	0.96	0.99	0.96–1.03	0.69

^#^ The hazard ratio refers to the subdistribution hazard ratio and not to the general hazard ratio. * Variables with *p*-value < 0.2 were selected from Table 1. ^†^
*p =* 0.054. CI: confidence interval; HR: hazard ratio; PA: Pseudomonas aeruginosa.

## Data Availability

Restrictions apply to the availability of these data. Data were obtained from Chiayi Chang Gung Memorial Hospital and are available from the authors with the permission of Chiayi Chang Gung Memorial Hospital.

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
