# Peer review of "Pseudomonas aeruginosa Infections Are Associated with Infection Recurrence in Arteriovenous Grafts Treated with Revision"

_medicina, 2023, doi:10.3390/medicina59071294_

Round 1

Reviewer 1 Report

The authors had a study on Pseudomonas aeruginosa infections associated with infection recurrence in arteriovenous grafts treated with revision. 

The subject is not novel. however, it is a main concern in its field as the total excision can be challenging due to the development of severe adhesions, and total excision can lead to bleeding, nerve injury, and arterial stenosis.

My comments:

The conclusion section of the abstract should be improved. It does not support the main results of the study. 

The introduction does not provide sufficient background and does not include all relevant references. It is also short. It must be improved and some recent and relevant references should be added.

The research design is appropriate. However, the subsections of the method section should be separated. Also, the ethical section should be transferred to the end of the manuscript. Besides, the trade name for all materials, instruments, and statistical software should be maintained.

Please improve the quality of Figure 2.

Why the authors used variables with p-values less than 0.2 ??

Why the authors excluded AVGs treated with total excision from further competing risk analyses?

The conclusion is so short and does not support the main results. Please improve.

Minor typo editing of English language is required.

Reviewer 2 Report

ABSTRACT

Background and objectives

Line # 19: following treatment with revision.

What does this mean?? please clarify.  

MATERIALS AND METHODS

Need paraphrasing and rearrangement of the knowledge in a consequence method.

Please report the type of the study (i.e. retrospective) as mentioned in the methodology section of the main manuscript, mention the criteria for patient selection (inclusion and exclusion criteria), the indications for AVG insertion, their follow up period, as well as reporting the timing of infection if present. 

Line # 20: the authors mentioned in the abstract section of the main manuscript the following: were performed on 60 patients to treat 65 AVG infections; but in the methodology section of the main manuscript lines # 65-66: Among the 60 patients enrolled in this study, 63 AVGs underwent surgical intervention for AVG infection, the number of patients is different. Please mention the correct patient’s number in both the abstract and the methodology section of the main manuscript.

Line 68-72: This patients underwent reinfection must be removed from the methodology section and reported in the result section of the main manuscript.

Please mention the type of the used graft, for patients that required revision, as well as the removed graft that are susceptible to pseudomonas infection. 

RESULTS

Did you performed culture and sensitivity test for the infected graft.

Please mention the timing of graft infection that required either revision or excision.

Reviewer 3 Report

Excellent article. The paper is well written in all its parts.

The authors should be congratulated.

I accept it in the current format.

Round 2

Reviewer 2 Report

Thank you for the authors for their revision and following the reviewers and editorial recommendations.